# Dynamically adaptive soft metamaterial for wearable human–machine interfaces

Ugur Tanriverdi[1,2,7], Guglielmo Senesi[1,2,7], Tarek Asfour[1,2,7], Hasan Kurt [1], Sabrina L. Smith[1,3], Diana Toderita[1], Joseph Shalhoub [4,5], Laura Burgess[6], Anthony M. J. Bull [1] & Firat Güder [1]✉

Our bodies continuously change their shape. Wearable devices made of hard materials, such as prosthetic limbs worn by millions of amputees every day, cannot adapt to fluctuations in the shape and volume of the body caused by daily activities, weight gain or muscle atrophy. We report a meta-material (Roliner) that is a dynamically adaptive human-machine interface for wearable devices. In this work, we focus on prosthetic limbs as the first application of Roliner. Roliner is made of silicone elastomers with embedded millifluidic channels that can be pneumatically pressurized. Roliner can reconfigure its material properties (behave like silicone or polyurethane with different shore hardness in different areas and times) and volume/shape based on the preference of the amputee in real-time, acting as a spatiotemporally adaptive meta-material. Preclinical studies of Roliner have demonstrated non-inferiority in operation and improved comfort for amputees.

The loss of limbs due to traumatic injuries or chronic diseases, such as diabetes or cancer, is an ever-growing healthcare problem: In the United States alone, there are currently more than 2.1 million people living with the loss of a limb, with the number of amputations expected to rise to 4.2 million by 2050, primarily due to diabetes[1,2].

To regain mobility or functions to carry out daily tasks, the majority of amputees require a prosthetic device, which is a wearable, load-bearing mechanical system. Although the invention and adoption of robotic limbs have greatly improved the experience of patients (for example, to walk or climb stairs more naturally), the issues surrounding poor fitting of prosthetics at the point of attachment to the body still need to be addressed[3]. The poor fitting of prosthetic limbs is a problem as old as the concept of prosthetics themselves and remains to be a substantial unmet medical need[3–10].

Modern prosthetic limbs are typically attached to the body of an amputee using a hard polymer or polymer-composite shell surrounding the limb with a soft sleeve (i.e., liner) in between to improve comfort. This shell is called the prosthetic socket. Prosthetic sockets are mostly custom-handmade for each amputee by a highly skilled technician through labor-intensive plaster casting methods using 19th century techniques[11]. Starting immediately after the amputation and throughout the lifetime of an amputee, the residual limb, primarily comprising soft tissue, continuously changes its shape for at least five reasons: (i) atrophy of the muscle mass because of the amputation; (ii) age-related anatomical changes; (iii) increases or decreases in body fat or water levels (for example retention due menstrual cycles or diet); (iv) post-surgery edema; and (v) exercise. Some of these changes in the shape and size of the residual limb occur over weeks to months (for example, due to atrophy soon after the amputation). Changes due to exercise, however, may occur over minutes to hours. In the current paradigm of medical practice, the long-term changes in the shape of a residual limb are remedied by manufacturing an entirely new custom socket, an expensive process; the short-term changes are still addressed primitively. It is left to the amputee to wear layers of socks over the liner throughout the day to adjust fitting, which is an ineffective, inconvenient, and short-term practice. The residual limb dynamically changes its shape, leading to a poor fitting. A prosthesis user wearing such an ill fit prosthesis may

[1]Department of Bioengineering, Imperial College London, London, UK. [2]Unhindr Ltd, London, UK. [3]Barts and the London School of Medicine and Dentistry, Queen Mary University of London, London, UK. [4]Imperial Vascular Unit, Imperial College Healthcare NHS Trust, St Mary's Hospital, London, UK. [5]Department of Surgery and Cancer, Imperial College London, London, UK. [6]Charing Cross Hospital, Imperial College Healthcare NHS Trust, London, UK. [7]These authors contributed equally: Ugur Tanriverdi, Guglielmo Senesi, Tarek Asfour. ✉e-mail: guder@ic.ac.uk

experience residual limb sores, ulcers, and possible further amputations[12].

To address the issue of changing limb geometry, current solutions have primarily focused on redesigning the socket. Over the past decade, companies such as Ottobock, Click Medical LLC, Martin Bionics and Limb Innovation Inc. have developed adjustable prefabricated or custom-made sockets to improve fitting[13,14]. These solutions, marketed by these companies globally with a major US presence, have straps or strings attached to the sockets, which alter fitting by increasing or decreasing the circumference of the socket. Although they offer an adjustable solution to fitting, they require the replacement of the entire socket with a new and expensive device costing over US $20,000 for the total replacement procedure. The high cost of these solutions is prohibitive in most regions across the globe. Some of these sockets also do not address the issues concerning the need for highly skilled technicians to measure, mold, and manufacture the initial socket, custom-made for each amputee. Perhaps the biggest issue with the adjustable socket solutions is that they only meet the needs for short-term (e.g., daily) modifications to fitting, such that if the geometry of the limb changes too much, an entirely new adjustable socket would need to be measured, molded and manufactured, costing another large sum. Furthermore, altering the socket geometry with string or strap-attached panels could alter load bearing, which is the primary function of a socket.

Because of the challenges surrounding adjustable sockets, a more effective and low-cost solution would be to use existing standard sockets available across the world with an adaptive, reconfigurable soft sleeve worn on the residual limb[15–18]. The soft adaptive sleeve (or liner) would go between the hard socket and soft residual limb and dynamically adjust its shape to compensate for the fluctuations in the volume of the residual limb. Currently, prosthetic liners sold commercially are not adaptive. Commercially available "passive" liners, made of silicone or polyurethane elastomers, are used by prosthetic users as a soft interface to improve comfort, suspension and the fitting but can not offer dynamic adjustment of these features. Prosthetic liners, however, can be turned into adaptive, soft robotic interfaces using methods borrowed from the emerging field of soft robotics, which allows the fabrication of soft actuators using fluidic systems.

In this work, we describe a reconfigurable and adaptive prosthetic liner to improve the fitting of prosthetic limbs (Fig. 1A). The adaptive liner, which we named "Roliner" (short for robotic liner), is made of low-cost silicone elastomers with embedded millifluidic structures that can be actuated electro-pneumatically. Roliner spatiotemporally reconfigures its shape/volume and material properties (e.g., shore hardness) to compensate for volume fluctuations of the residual limb to improve fitting and comfort dynamically. We extensively characterized the material properties (such as tensile, compressive and volumetric elasticity) of this new class of liners and produced a miniaturized, battery-operated electronic control unit and cloud-enabled smartphone application to operate the liners on demand by prosthesis users. We validated the Roliner (including the control unit and smartphone application) in pre-clinical human trials and compared to standard "passive" liners to demonstrate the effectiveness of Roliner as a dynamically reconfigurable metamaterial to improve fitting.

## Results
### Design and fabrication of Roliner
We envisioned Roliner as an adaptive millifluidic-based prosthetic liner with the following design features: Roliner must support (i) reconfigurable mechanical properties; (ii) electro-pneumatic controls to enable the use of embedded millifluidic air channels; and (iii) instant personalization and monitoring over cloud computing as shown in Fig. 1A.

An essential characteristic of prosthetic liners is to be thin enough to instill a heightened sense of prosthetic control amongst amputees. Most commercially available "standard" liners have a tapered profile with a typical thickness below 6 mm[19,20]. Creating an adaptive prosthetic liner with embedded channels within this thickness limit, while being compatible with high-volume manufacturing and capable of withstanding high levels of mechanical stress, particularly when pressurized, is a great challenge. The existing fabrication methods for prototyping, such as 3D printing or microfabrication, are slow and unfeasible to produce micro or millifluidic channels in elastomers at scale, thus rendering them incompatible with industrial high-volume manufacturing[15,16].

We have developed a new fabrication technology (Fig. 1B) to produce Roliner with embedded millifluidic channels that can be pressurized up to 30 psi while being compatible with high-volume manufacturing. In this scheme, a 1 mm thick initial silicone layer was molded onto an acrylic frame. To produce the embedded fluidic network of channels on top of the initial silicone layer, we exploited the chemically inert nature of silicones, preventing bonding with other interlayer materials[21]. Initially, patterns were digitally designed using a vector graphics software and uploaded onto an automatic cutter-plotter to recreate the desired geometry cut into a sheet of food-grade wax paper with a nominal thickness of 250 μm. Next, we placed the patterned wax paper onto the initial silicone layer and semi-cured both at room temperature for 25 min to ensure a gel-like consistency. Semi-curing of the silicone layer is imperative for two important reasons: (i) to improve the bonding of subsequent layers of silicones to the initial layer; and (ii) to create a solid enough surface to suspend the additional layer of wax paper. We poured another layer of uncured silicone mixture to embed the wax paper and cured the assembly at room temperature for 4 h to form embedded millifluidic channels. In essence, the patterned wax sheet prevented the chemical bonding of

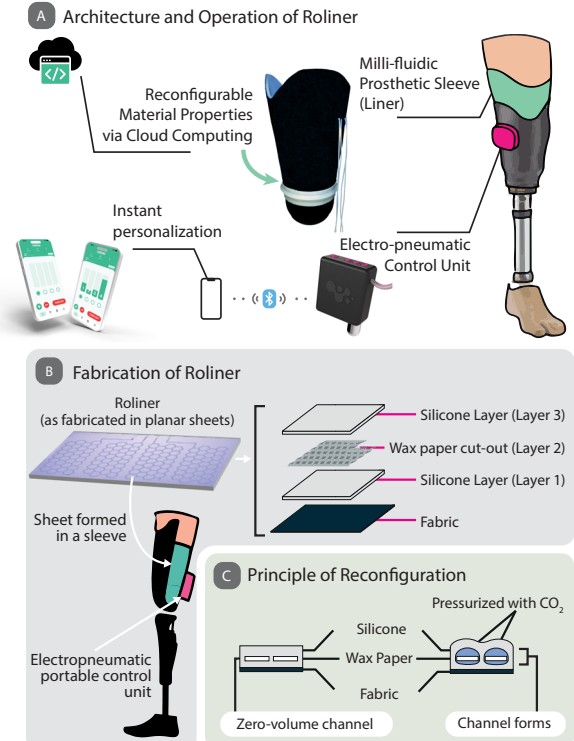

**Fig. 1 | Fabrication of Roliner and working principle are shown in a prosthetic limb setup. A** The schematic illustration of the fabrication of the Roliner and photograph of the actual Roliner. **B** Roliner, as a dynamic interface technology, is shown in prosthetic limb application: The control unit and liner components of Roliner are shown in an above-knee prosthetic setup. The layers are shown in the exploded view. **C** A fluidics architecture that is flat when deflated and only increases in thickness when pressurized.

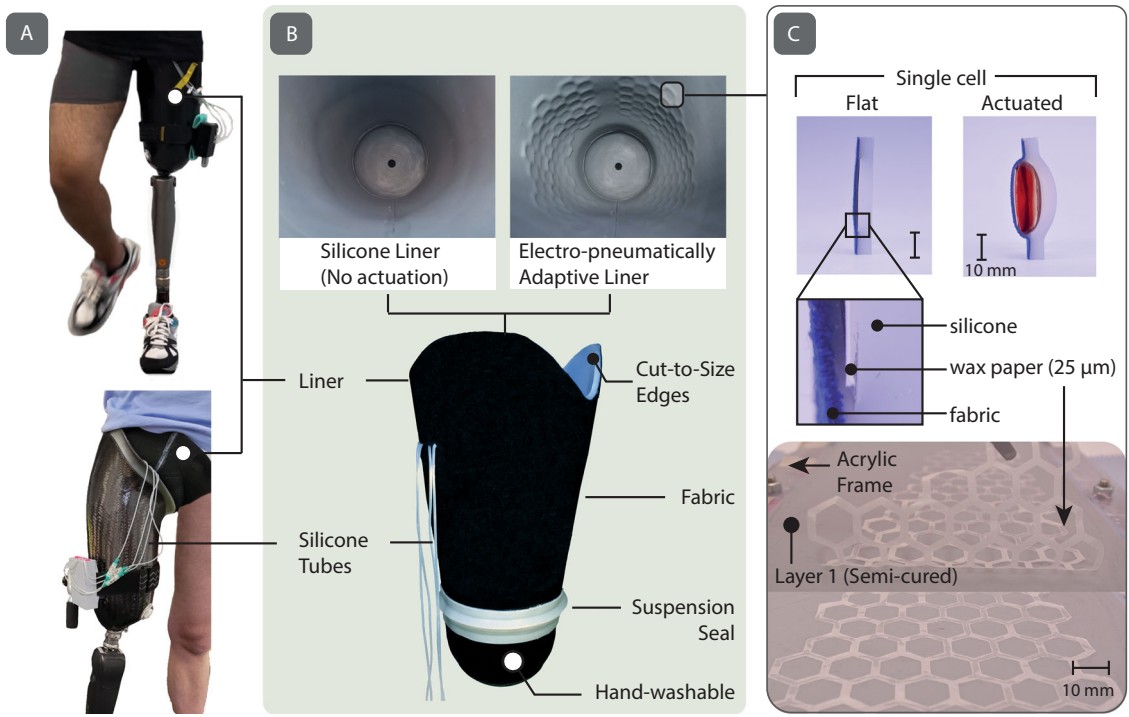

**Fig. 2 | Roliner in operation on transfemoral amputees. A** The outer and inner features of Roliner. **B** The pressure actuation demonstration of millifluidic-embedded single cells of Roliner (**C** top) The fabrication of millifluidic channels on Layer 1 (**C** bottom).

the top and bottom silicone (Layers 1 and 3 in Fig. 1B), acting as an area-selective mask. The embedded millifluidic channels formed were fully planar with a nominal thickness of less than 250 μm when not pressurized—i.e., near-zero volume (Fig. 1C). When pressurized (e.g., electro-pneumatically), the embedded planar millifluidic channels could expand in volume over 100 times, essentially transforming into a 3D structure.

To form a prosthetic liner that can be worn by the prosthesis user (Fig. 2), we rolled the planar silicone sheets with the embedded channels to join both ends to form a cylinder. A thin layer of silicone adhesive was utilized to permanently attach the long edge of the sheet together (Fig. S1). Subsequently, we attached an additional dome-shaped piece of silicone (molded previously and also cured at room temperature for 4 h; see Fig. S2) to support the distal end of the residual limb. When pressurized, the millifluidic channels embedded in the liner can expand omnidirectionally or

unidirectionally depending on presence of a strain-limiting layer (e.g., woven fabric) adhered to the silicone layer (Fig. 2C, Layer 1). The strain-limiting layer can also provide additional structural reinforcement to prevent uncontrolled expansion of the liner due to the snap-through instability of the relatively thin elastomer[22]. The silicone sleeve was, therefore attached to a fabric (Domyos, Decathlon UK) using silicone adhesive. The fabric coating also reduced the tackiness of the surface of the liner and made the donning of the prosthesis easier for the prosthesis wearer. Finally, we placed a silicone suspension ring on the outer edge of the liner to achieve suspension while walking and attached four silicone tubes for the pneumatic actuation of the four independent zones of Roliner (See SI for alternative manufacturing methods for Roliner).

## Investigation of the material properties of the reconfigurable composite

Unlike bulk elastomers, rubbers with embedded fluidics can be classified as composite materials[23]. By modulating the pressure of the fluid, the ratio of the mass between the elastomer and fluid can be actively modified, resulting in tunable mechanical characteristics, in essence,

an entirely new material—metamaterial[24–26]. The mechanical characteristics of the overall composite material can be *configured* through the precise pressurization of the millifluidic channels, leading to a dynamically reconfigurable material. To investigate how the pressure of the fluid within the channels modulates mechanical characteristics, we investigated the tensile (TE), compressive (CE), and volumetric (VE) elasticity of the dynamically reconfigurable composite over a range of levels of fluidic pressures.

To determine the TE characteristics of the reconfigurable composite (Fig. 3 (top)), we molded a silicone specimen with an embedded linear fluidic channel and a polyurethane specimen (width of 3 mm and unactuated channel height of ~250 μm) in the shape of a standardized dogbone (ASTM D412-16)[27]. We measured the tensile stress at 0–100% strain range at three different actuation pressures (0, 10, and 15 psi) using a material testing machine (MTM, MultiTest 5-xt, Mecmesin, UK). Next, we measured the tensile modulus of the reconfigurable material at different actuation pressure levels between the range of 5–20% strain range (*n* = 3). The stretchability of the composite material can be dynamically tuned purely by either increasing or decreasing the pressure within the fluidic channels. A material with reconfigurable tunable TE would allow amputees to modulate the stretchability of the liner dynamically (unlike "passive" liners) depending on their activity, such as walking or sitting, or preference since the amount of stretch required in the liner is generally variable.

We characterized the CE of the reconfigurable composite Fig. 3 (middle). We molded silicone and polyurethane (PU) specimens with an embedded fluidic channel (lateral hexagonal channel area of 3 or 6 cm² and un-actuated channel height of ~250 μm) with a silicone tube connected to the channel from the side. To ensure precise CE measurements, we supplied a constant actuation pressure to reduce the backflow of the fluid in the channel. Unlike the TE experiments, the specimen shape was a disc, with a diameter of 45 mm and an average thickness of 6.63 mm (Table S1). We first performed a specimen pre-conditioning procedure to ensure accurate measurement of CE since the viscoelastic properties of elastomers under stress change over time[28]. For the preconditioning procedure, we applied a cyclical

## Multi-mode Material Analysis of the Reconfigurable Composite

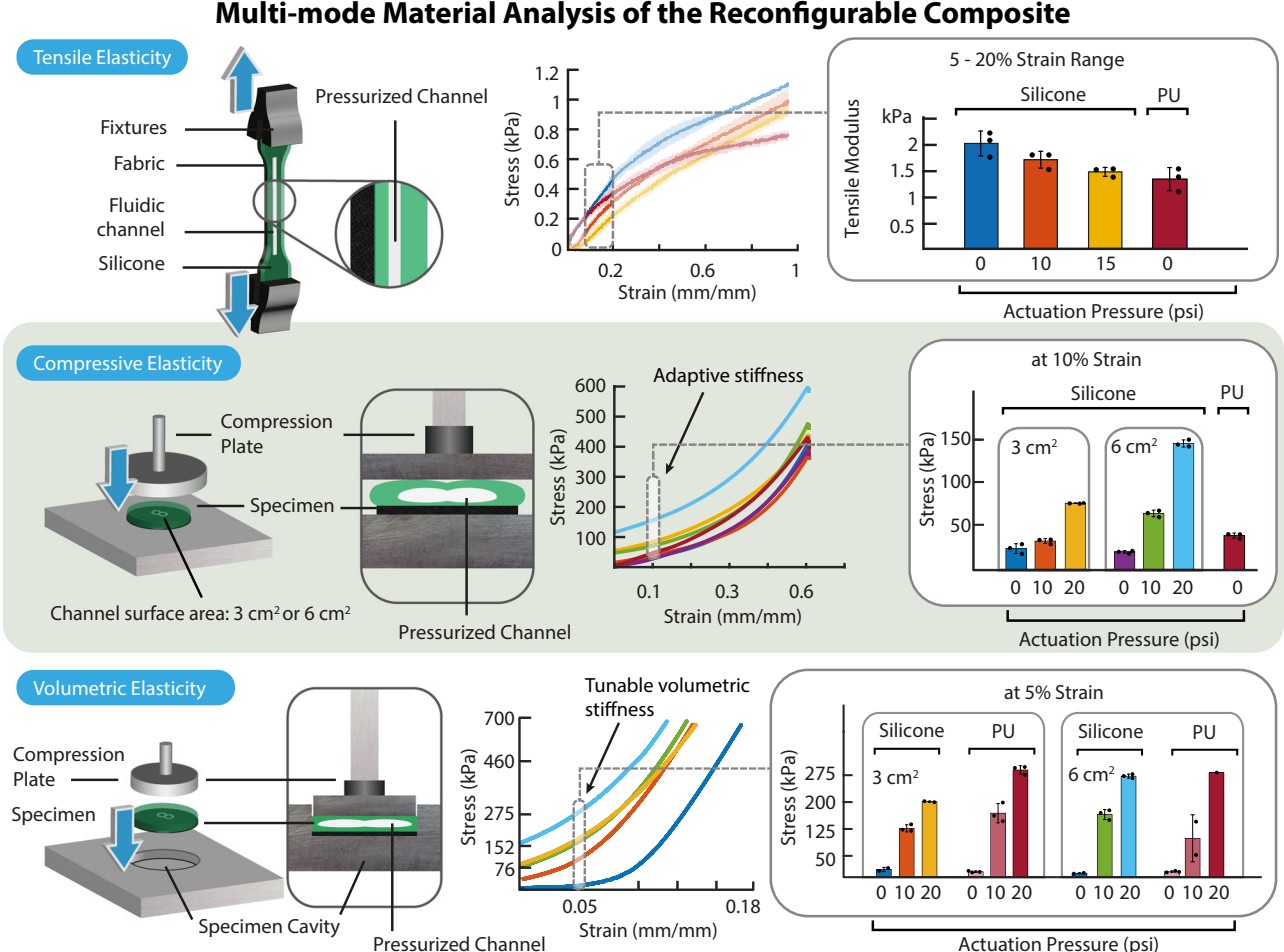

**Fig. 3 | Multi-mode material analysis of the reconfigurable composite.** Tensile Elasticity experiment results (**top**). From left to right: TE experiment setup, stress vs strain graph with 5%–20% strain region highlighted, tensile modulus (slope) between 5 and 20% strain range. Data are presented as mean values and the error bars where shaded regions represent the standard deviation (± σ). Compressive Elasticity results (**middle**). From left to right: CE experiment setup in 3D illustration and the cross-section, stress vs strain graph of all specimens, with 10% strain highlighted, stress vs pressure graph of specimens at 10% strain. Volumetric Elasticity results (**bottom**). From left to right: VE experiment setup in 3D illustration and the cross-section, stress vs strain graph of silicone specimens with 5% strain highlighted, stress vs pressure graph of specimens at 5% strain. The data in bar charts are presented as mean values ± SD where applicable. The individual data points of each independent experiment ($n = 3$) were presented as overlayed on the bar charts.

compressive stress of 250 kPa at a strain rate of 100% per second (equivalent to 398 mm·min$^{-1}$) 550 times. We estimated the CE of each preconditioned specimen at

10% strain ($n = 3$). We actuated the channel at pressures of 0, 10, and 20 psi in compression mode using the MTM. We found that the CE of the reconfigurable composite material increased proportionally to the actuation pressure as we increased the ratio (by mass) of the compressible fluid ($CO_2$ gas) to the nearly incompressible material (silicone elastomer). Through actuation, our silicone-based reconfigurable composite material can match and surpass the mechanical characteristics of PU as shown in Fig. 3 (middle). In the context of prosthetic liners, CE is an important material characteristic governing the comfort and control of the limb. CE defines the ability of the liner to distribute the ambulatory stresses (comfort) and to stabilize the mechanical coupling between the limb and socket (control). While a harder liner (high CE) improves control and offers lesser comfort, a softer liner (low CE) distributes peak pressures and provides cushioning, making the liner more comfortable. Using the dynamically reconfigurable composite material, the amputee can modify the CE characteristics of the liner such that the control vs comfort trade-off can be modulated depending on the activity (i.e., sitting or walking).

The degree of stiffness can be increased either by increasing the pressure or the size of the fluidic channel (3 cm² vs 6 cm²–Fig. 3 (middle)) allowing several degrees of variables to modulate the mechanical characteristics of the material.

The tensile, compressive, and volumetric elasticity of the dynamically reconfigurable composite material provides a wide range of control over the stretchability, comfort/control, and adaptability of a prosthetic liner, respectively.

**Characterization of actuation and suspension force of Roliner**
When fabricated into a wearable sleeve, the dynamically reconfigurable composite plays two critical roles as a prosthetic robotic liner: (i) increasing comfort/fitting; and (ii) improving the suspension of the artificial limb. While the amputee is in motion, adequate suspension force needs to be supplied to ensure the suspension of the artificial limb to prevent falls and slippage (also known as pistoning). While at rest, however, the suspension force can be reduced for comfort.

In essence, Roliner acts as an array of radially distributed soft actuators and exerts an inward actuation force ($F_A$) normal to the liner surface when pressurized, as shown in Fig. 4A. When Roliner is worn around the residual limb, the outward forces exerted ($F_A$) increase the

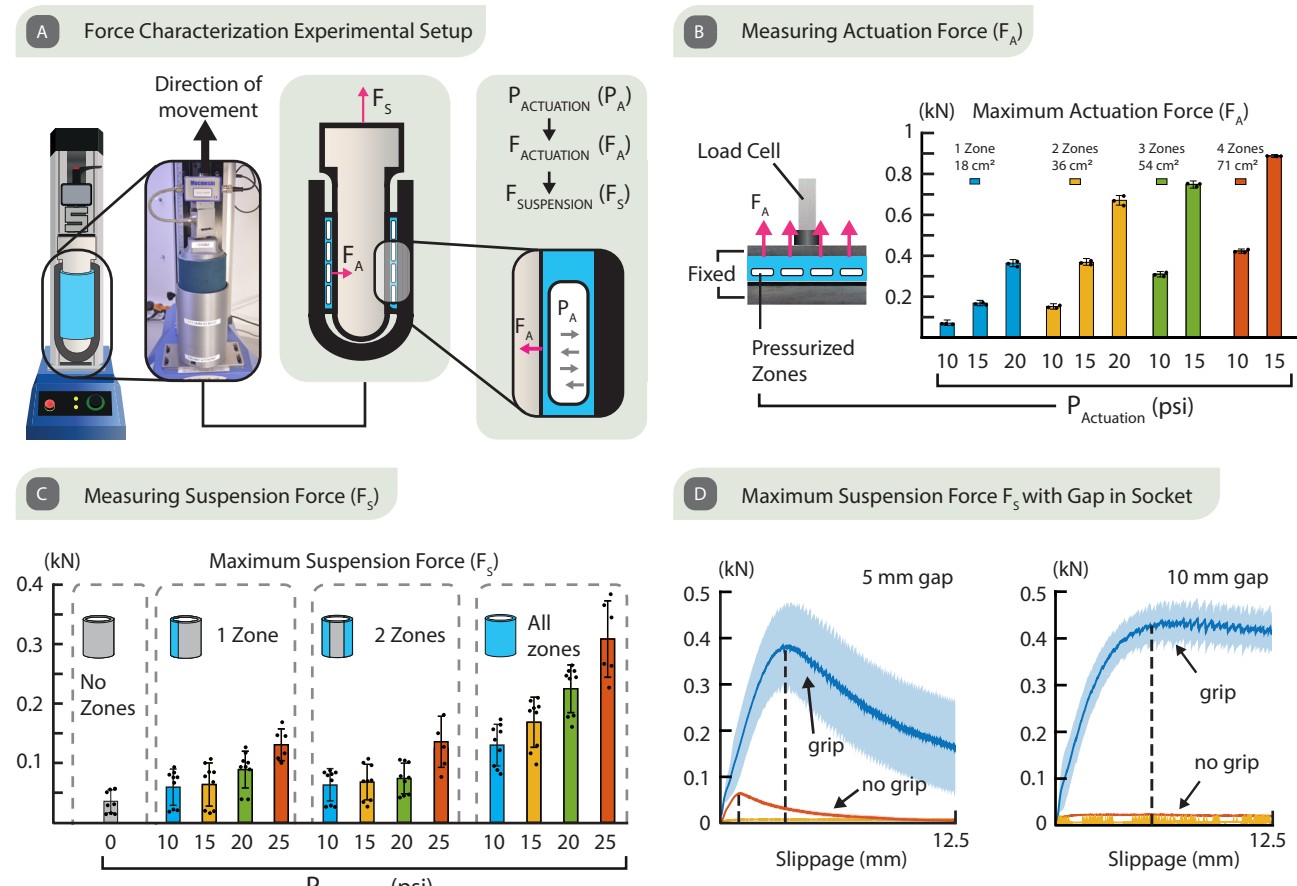

**Fig. 4 | The mechanical characterization of Roliner. A** The measurement setup for the investigation of actuation pressure ($P_A$), actuation force ($F_A$), and suspension force ($F_S$). **B** Measuring Actuation Force: measurement setup with a horizontally restrained flat liner (left). The maximum actuation forces observed at differing levels of actuation pressure ($P_A$) and with 1–4 zones of actuation are shown (right). Data are presented as mean values ± SD. The individual data points of each independent experiment ($n = 3$) were presented as overlayed on the bar charts.

**C** Measuring Suspension Force ($F_S$) observed with differing levels of actuation pressure (PA) with 1, 2, and 4 zones of actuation are shown. Data are presented as mean values ± SD. The individual data points ($n = 9$) of each independent experiment ($n_i = 3$) and technical repeats ($n_t = 3$) were presented as overlayed on the bar charts. **D** The maximum suspension force ($F_S$) vs displacement observed in the sockets with a gap of 5 mm and 10 mm. Data are presented as mean values and the error bars where shaded regions represent the standard deviation ($\pm \sigma$).

friction between the liner, residual limb, and socket. The increased friction results in a greater suspension force ($F_S$) in addition to other suspension methods thus ensuring the better attachment of the residual limb to the socket, which is particularly important when the amputee is in motion.

To investigate the relationship between $F_A$ and the actuation pressure ($P_A$), we constrained a planar version of Roliner, which has four individual zones of embedded millifluidic channels, between two flat metal plates. Then, we measured $F_A$ at different $P_A$ levels (10–20 psi) and increasing number of pressurized zones (up to 4 zones) using an MTM (Fig. 4B). We observed an $F_A$ of up to 350 N by pressurizing a single zone (with an area of 18 cm²) of Roliner. It is possible to achieve $F_A$ above 1000 N with an increased number of actuated zones and actuation pressure. However, we were limited by the upper measurement limit of MTM, so we refrained from applying $P_A$ of 20 psi in the measurements of Roliner with 3 and 4 actuated zones. Roliner can reproducibly achieve an $F_A$ of 900 N when all four zones (71 cm²) are actuated at $P_A$ of 15 psi. These results underline the fact that it is possible to generate a significant amount of $F_A$ using electro-pneumatically actuated reconfigurable composites.

Next, we investigated how much suspension force ($F_S$) can be exerted by the cylindrical form of Roliner depending on the level of actuation pressure (Fig. 4C). For this experiment, we used an impression made of plaster from an amputee which we acquired from the

Douglas Bader Rehabilitation Facility at St. George's University Hospital National Health Service Foundation Trust in London, UK. To simulate soft tissue, the impression was wrapped with a 40 mm thick foam rubber (IdealTek ESD Foam, UK). We used a Roliner with four actuation zones but without the bottom end cap. Roliner was placed on the foam rubber wrapped plaster impression and inserted into a 3D-printed prosthetic socket made of polylactic acid. We fixed the socket onto the bottom plate of the MTM and pulled the impression upward under a range of $P_A$ (0–25 psi) within four zones of actuators of Roliner as shown in Fig. 4A. By increasing $P_A$ and modulating the actuation configuration of zones within Roliner, we were able to considerably increase $F_S$ above 300 N, compared to ~35 N without actuation. We also determined that a 5 mm thick Roliner could expand and fill a gap of up to 10 mm between the socket and the leg, as shown in Fig. 4D; such a gap of 10 mm would ordinarily render walking impossible.

## Portable electro-pneumatic control unit

Roliner requires a mobile pressure management system. For the operational deployment of such a system, it is imperative to carefully consider the mobility, usability, and fault-tolerance features.

Typically, electro-pneumatic systems operating at relatively large pressures (> 20 psi) pressurize the fluid with power-hungry motor-driven pumps and require large valves and manifolds for the manipulation of fluids[29–31]. To meet our high mobility requirement, we have

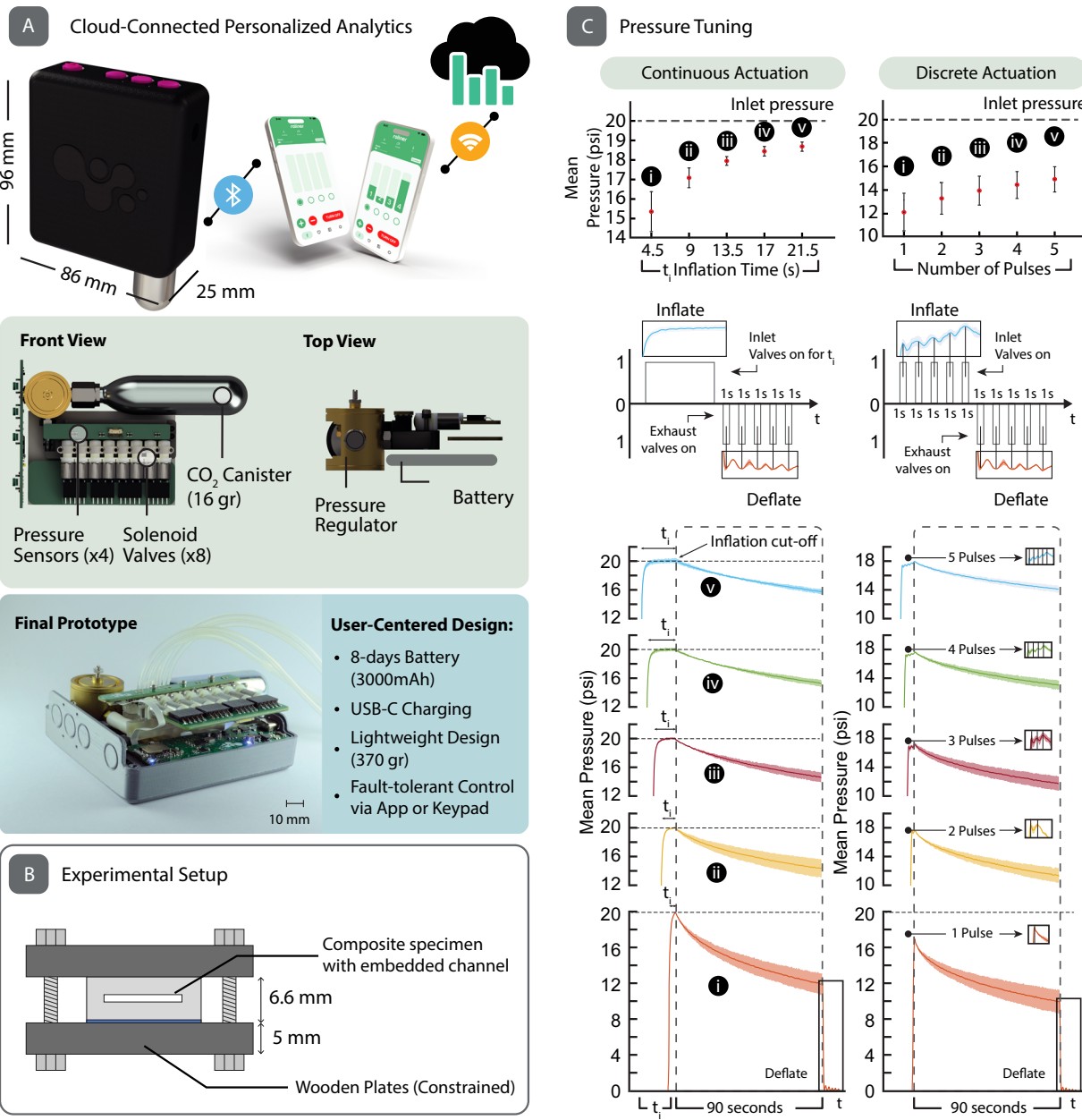

**Fig. 5 | Actuation control unit and pressure tuning of Roliner. A** The rendered concept figures of the control unit along with its dimensions and cloud connectivity (top). The rendered technical figures of the control unit from front and top views (middle). The real final prototype and operational specifications. The scale bar shows 10 mm. **B** Experimental setup used in all experiments shown in (**C**). **C** The pressure tuning of the electro-pneumatic control unit. The upper panel (the first row of plots) shows the mean pressure ($n = 3$) reached with respect to the inflation time in both continuous (left) and discrete actuation (right). Data are presented as

mean values ± SD. Actuation profiles for the inlet and exhaust valves are shown along the pressure profiles during deflation in the second row. Data are presented as mean values. The shaded regions represent the standard deviation ($\pm\sigma$). The third row displays the mean pressure profiles ($n = 3$) of the millifluidic channels during continuous and discrete pulse actuation (up to 5 cycles) over 90 s. Data are presented as mean values. The shaded regions represent the standard deviation ($\pm\sigma$).

---

designed a fault-tolerant, small (~ 210 cm3) and light (~ 370 gr) battery-powered control unit which can be easily attached to the socket to inflate/deflate the embedded fluidic channels within Roliner (Fig. 5A). The control unit consisted of eight custom-designed electro-pneumatic valves (HDA0531115H, The Lee Company, USA), which were electrically controlled by a Bluetooth® enabled, System-on-a-Chip wireless microcontroller (NRF52832, Nordic Semiconductor ASA, Norway). A pair of valves were utilized to modulate the fluidic pressures within each individual zone of Roliner. These valves were surface mounted on a 3D-printed transparent manifold. To monitor the pressure of the fluid inside the channels, the actuation pressure of each

individual zone was monitored using a solid-state pressure sensor (ABPDANN060PGAA5, Honeywell, USA) in the pressure range of 0–30 psi. To avoid using a large and heavy battery, we employed a miniature pressurized gas cylinder containing 16 gr of $CO_2$ instead of a diaphragm pump. These commercially available gas cylinders are pressurized to ~900 psi (at 22 °C) and commonly used by cyclists and in emergency lifejackets for rapid inflation, hence they are low-cost. To downregulate the high-pressure of the gas cylinder to the actuation pressure of Roliner (0–30 psi), we used a custom-made pressure regulator (Beswick Engineering, USA). Based on battery usage by the amputees in the pre-clinical experiments, we determined that the

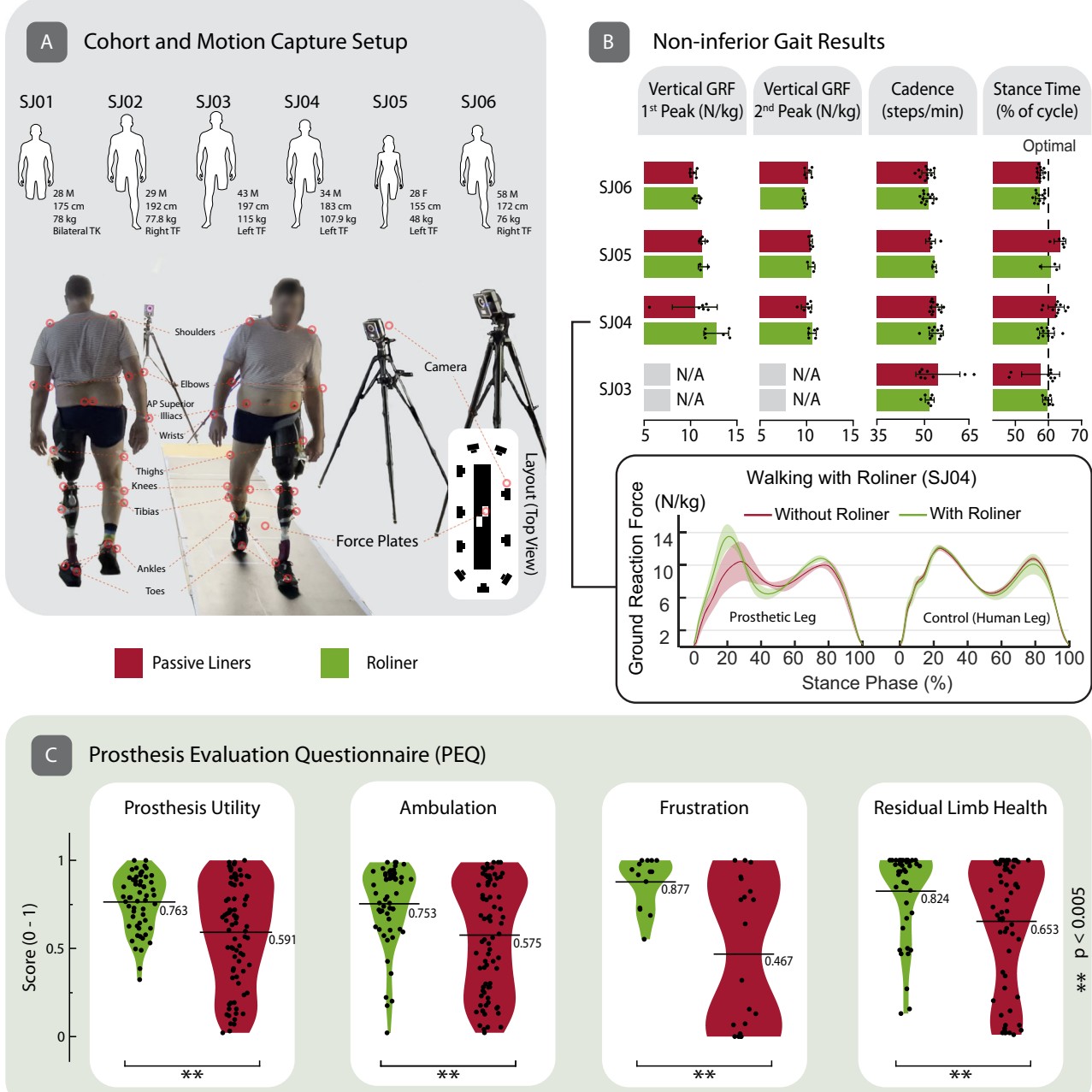

**Fig. 6 | Gait cycle analysis and prosthesis evaluation questionnaire results.**
**A** Cohort description and SJ03 walking in the motion capture setup. (TK: through the knee, TF: transfemoral) **B** First and second peaks of vertical ground reaction force (GRF) of the prosthetic leg, cadence, and stance time of the subjects. The detailed GRF vs stance phase of SJ04 who walked with a poor-fitting socket because of 20 kg weight loss. Data are presented as mean values ± SD. The individual data points of each independent experiment ($n \geq 5$) were presented as overlayed on the bar charts. **C** The violin chart comparison of Roliner versus passive liners in terms of prosthetic utility, ambulation, frustration and residual

limb health parameters Prosthesis Evaluation Questionnaire (PEQ) scores from the preclinical cohort. The horizontal line shows the mean score. The mean scores ($\mu$) are shown on the right-hand side of the violin chart. In the violin charts, kernel density estimation was performed using Scott type bandwidth. The individual data points represents the independent PEQ scores from the related section ($8 \geq n \geq 2$) of the questionnaire for each participant ($n = 6$). These were presented as overlayed on the box charts. The $p$ value calculated using two-sided paired sample $t$-test. (**) represents statistical significance as $p < 0.005$.

control unit could last up to eight days for continuous use with a battery capacity of 3000 mAh.

In the pre-clinical studies, we observed that the amputees, once they put on the liner prior to walking, interacted with the device in two distinct modes: the amputees either (i) continuously pressed and held the actuation button on the mobile application to inflate the liner; or (ii) pressed in quick succession multiple times to inflate the liner. To understand the relationship between the actuations characteristics

and pressurization of the liner, we carried out a series of experiments. We constrained a planar specimen of a single zone from a Roliner between two wooden plates to simulate the volume constraining environment for a single actuator within the prosthetic socket (Fig. 5B). In all these experiments (Fig. 5C), the specimen was pressure-cycled between actuation pressure to atmospheric pressure with a hold duration of 90 s each cycle, which is approximately the period before initiating movement by an amputee. When the specimen was

actuated continuously for durations between 4.5 and 21.5 s (multiples of 4.5 s), Roliner quickly reached the rated inlet pressure of 20 psi. Once pressurization stopped, over the next 90 s, the pressure within Roliner dropped to a level relative in magnitude to the duration of actuation (Mean Pressure). This drop in pressure is attributed to the lateral expansion of the elastomer, increasing the volume of the channels. Since longer actuation introduced more mass (in the form of $CO_2$ gas) in the channels, the mean pressure was always higher at the end of the 90 s hold period. In another experiment, the channels were pressurized with shorter quick pulses of up to one second with one second separation between each pulse, up to five consecutive pulses, instead of long and continuous pulses. After comparing continuous and pulsed actuation, the most important outcome is that although the initial peak pressure is lower for pulsed pressurization (18 vs 20 psi), the mean pressure within the channels is the same over the 90 s hold period (see five pulses (5 s) actuation vs 4.5 s continuous actuation). We believe that the initial lower pressure during pulsed actuation originates from the fact that the pauses between pulses offer the elastomer enough time to expand, thus resulting in lower peak pressures. Amputees who continuously actuate Roliner will likely first experience tightness, which may lead them to deflate the liner unnecessarily. This experience can be avoided by pulsed actuation, which would also save pressurization gas from the portable canister, allowing longer-lasting usage up to 2000 actuations.

### Validation of Roliner in pre-clinical human trials

In order to translate a medical technology from the laboratory setting to clinical use, non-inferiority of the medical device in question should be demonstrated against a similar clinically "gold-standard" device[32,33]. For prosthetic technologies, non-inferiority can be supported by measuring the objective and subjective performance of the device in clinically relevant settings. Only if the investigational (i.e., Roliner) is non-inferior to the clinical gold-standard (in this case an existing "passive" liner), can it progress within the regulatory process to receive medical approval for use in healthcare systems.

We validated Roliner along with both accompanying hardware and software in a set of pre-clinical human trials with six amputees from diverse (e.g., age, ethnicity, sex, height, weight, etc.) backgrounds (Fig. 6A), who provided written informed consent. The volunteer amputees were invited to participate in an experiment to evaluate the performance of Roliner both objectively—by gait analysis using a motion capture system—and subjectively—through participant reported outcome measures (PROMs). These two techniques are routinely used for the evaluation of clinical efficacy. All participants were unilateral (i.e., absence of a single leg) above-knee amputees except for SJ01, who was a bilateral through-knee amputee. One of the participants was female, one was above fifty years of age, and two were people of color. The height and body mass of the participants ranged from 155 to 197 cm and 48 to 115 kg, respectively. To measure operational performance, the volunteers were asked to wear Roliner with their everyday prosthetic limbs (including their sockets) and operate Roliner using the smartphone app to their preference. For comparative analysis, each experiment was repeated with both Roliner and a conventional "passive" liner that is clinically used (i.e., the gold standard).

In the first experiment, we aimed to understand the objective differences through gait analysis between Roliner and clinically used "passive" liners. For this, the participants ($n = 4$) were asked to walk on a 6-meter-long platform. This was repeated a total of six times for each liner. For gait analysis, we used a motion capture system (Vicon Motion Systems Ltd, UK) which consisted of ten infrared (IR) cameras placed around the platform walkway which tracked 16 wearable IR reflective markers. The walkway was also equipped with two force plates placed in the middle to measure the vertical ground reaction force (GRF) of each foot during walking. Using the position of the reflective markers in 3D space and signals obtained from the force plates, we estimated a range of metrics related to gait including: stance/swing time, vertical GRF, cadence, stride length/width, range of motion (for ankle, hip, knee joints) and self-selected speed (please see supplementary information for a detailed description of these metrics). This objective analysis demonstrates that Roliner did not impede the natural gait of the participants in comparison to their usual clinically prescribed liner. Roliner was shown to improve various aspects of gait in a number of the objective metrics evaluated. For example, stance time, which is defined as the time spent on a foot in a cycle of heel-to-heel contact, should be 60%–40% in able-bodied individuals. We have observed that stance time improved in three of the four participants. With these objective evaluation results, we have shown that Roliner is not inferior to the clinical gold standard in terms of gait with the added benefit of reconfigurable fitting (Fig. 6B and Tables S3–6).

All limbs are geometrically unique, and every individual has a different pain threshold and preferences for comfort—e.g., harder vs softer liner. To evaluate the subjective experience of Roliner, we asked the amputees to complete a disease-specific validated questionnaire, also known as the prosthetics evaluation questionnaire (PEQ, see supplementary information). PEQ is a clinically accepted standard for PROMs that measures the subjective experience of amputees with new prosthetic devices and liners. PEQ evaluates a prosthetic device across nine domains; however, only four of these nine are relevant for the evaluation of Roliner: prosthetic utility, ambulation, frustration and residual limb health. As shown in Fig. 6C, we have observed statistically significant improvements in the outcome measures with a lower spread in all four domains (Tables S9–20). All amputees found Roliner to be more effective as a liner. For example, all participants found Roliner to improve fitting and their experience to be less frustrating compared to their everyday liner.

## Discussion

Roliner is a reconfigurable, soft metamaterial composite based robotic liner that works as a fitting interface, compatible with existing prosthetic sockets—carbon fiber composite, thermoplastics, etc. Although Roliner is made of silicone elastomer only, the mechanical properties of Roliner can be dynamically altered to behave like another type of elastomer, such as PU, thanks to its reconfigurable metamaterial composite design.

Unlike conventional prosthetic sockets, Roliner does not require custom handmade manufacturing by a specialized technician, often not easily accessible worldwide. Roliner extends the effective lifetime of prosthetic sockets by dynamically improving fitting with a simple touch of a button. For example, subject SJ04 had rapid weight loss of 20 kg prior to our pre-clinical studies. He did not have a socket replacement after losing weight therefore, he consistently experienced poor fitting everyday with his prosthetic socket. The gap between his residual limb and the socket was in the order of 1.5 cm, and the subject manually filled this gap by wearing additional layers of ply socks to walk. Once the subject wore Roliner, the fitting problem was immediately solved (Fig. 6B): When actuated, the subject could comfortably move around with confidence within seconds. In fact, Roliner rendered an otherwise unusable prosthetic socket usable, thus extending the usable lifetime of the socket with a simple touch of a button.

Both Roliner and the actuation control unit (and accompanying mobile app) were designed and produced in our laboratory using commonly available methods such as 3D printing, laser cutting, and degassing. Hence, we expect that these devices can be prototyped easily in most academic settings. Roliner is made of medical grade materials and ready for certification (developed in line with ISO13485, IEC 60601, ISO 10993, ISO 11607, ISO 14155, ISO 14971, ISO 15223, ISO 62304, ISO 62366) to be used in health systems in Europe, USA and rest of the major markets[34–36]. Furthermore, Roliner is compatible with high-volume automated manufacturing technologies yet washable and

cut-to-size, which was part of our design philosophy from the conception of the idea. Currently, most prosthetic sockets are handmade in clinics with long wait times (3–4 weeks in UK and US); hence, most amputees experience poor prosthetic fitting in their daily routines as their limb shape would change in this duration.

Roliner has the following issues that need to be addressed to accelerate worldwide adoption: i) The total bill of materials for producing a single unit of Roliner prototype is the order of US $50 for small batch production. We expect a reduction (> 50%) in materials cost for larger volume manufacturing (>1000 units). ii) The portable electro-pneumatic control unit prototype currently costs around US $700 per unit in the laboratory setting. We also anticipate at least a twofold reduction in the unit cost at larger volumes, reaching -US $350 for a manufacturing volume of ~1000 units. The control unit also requires the use of $CO_2$ cartridges as consumables, adding a running cost to the overall Roliner assembly. $CO_2$ cartridges cost as little as US $0.86 and can last over a week. iii) The preclinical study did not have a large cohort of amputees and sufficient duration to draw statistically sound conclusions. A more extensive study with a larger cohort of participants is required to definitively evaluate the clinical efficacy of Roliner, likely through a long-term crossover study (i.e., amputees need to wear their liner for a longer period and then switch over to Roliner).

The initial cost of Roliner will be higher than that of a "passive" liner. It will, however, provide cost savings by reducing socket replacements and productivity losses due to the time spent in clinics fitting new sockets. Most importantly, Roliner will substantially enhance the quality of life for amputees as the main added value.

In the future, the reconfigurable soft metamaterial composite can be integrated with flexible electrodes to perform non-invasive continuous monitoring of both biophysical and biochemical targets for wearable applications. Some intriguing examples of such integration would be electrocardiography, electromyography, and biosensing of various biochemical analytes in sweat (such as sodium or glucose levels)[21,37–44]. Implementation of electrode arrays for electromyography may be especially useful for high-precision control of robotic prosthetics, which is an area of active research[45,46]. Although we used the reconfigurable, soft metamaterial composite as a prosthetic liner in this work, the reconfigurable metamaterial composite proposed will likely find other applications where the soft human tissues are interfaced with rigid wearable materials such as knee braces, ski boots, and exoskeletons.

## Methods

### Fabrication of Roliner
We designed a hexagonal fluidic pattern on Adobe Illustrator and cut the pattern on wax paper by using a plotter cutter (Graphtec CE7000 Series, UK). The layout was then placed on 1st layer of semi-cured silicone 25 min after pouring the 1st layer at 22 °C. After placing the pattern on semi-cured silicone, we poured the 2nd layer of uncured silicone until it reached the rim of the mold (5 mm). Once the silicone layers were cured, the wax paper formed channels with a thickness of 0.25 mm, behaving as a placeholder during curing (Fig. 2). After curing, thanks to the texture and softness of the wax paper, the embedded wax paper in the channels would not hinder the pathways to function, even if the wax paper had been disintegrated.

After demolding the flat, flexible sheet, we merged the ends of the sheet to form a cylinder using the silicone-epoxy glue Silpoxy® (by Smooth-On Inc., PA, USA). Once dried, a silicone cap (dome) made of Ecoflex OO50 was glued to the distal end of the cylinder, forming the liner's characteristic U-shape.

Through pierced holes, silicone tubes (Hose Pipe Tubing ID: 0.5 mm, OD, 2.1 mm by Advanced Fluidic Solutions, UK) were attached to the beginning of the channels with Silpoxy® adhesive. Lastly, the outside of the silicone liner was smeared with Silpoxy glue, and the fabric sock was rolled on from bottom to top.

The methodology presented here can be extended to various manufacturing processes, such as injecting polymers into cavities or molds, as opposed to the presented technique of pouring polymers into flat molds. Furthermore, chemical processes characterized at specific pressure and temperature conditions, such as plasma bonding or the use of adhesives, can be employed to bond two cured or semi-cured geometries made of polymers. Alternatively, encapsulating a flat polymer sheet with embedded channels by using over-molding and injection-molding techniques can also achieve a high-volume manufacturing output. These alternative methods could still result in the encapsulation of flexible support materials, such as wax paper or other flexible materials formed as sheets, or pouches, to create channels within the polymers.

### Materials Characterization
**Tensile Elasticity:** Two sets of 12 dog-bone specimens with fabric backing were fabricated to observe and characterize Roliner's tensile elasticity, following ASTM D214 standard (American Society for Testing and Materials, - ASTM, 2016). The specimens contain a 70 × 2.8 mm wax paper channel in the narrow section, where the channel was connected to a silicone tube for actuation. The specimens were fixed on the top and bottom grippers of the MTM (Mecmesin Multitest 5-xt, UK) and actuated to 0, 10, and 15 psi.

**Compressive Elasticity:** The specimens were fabric-backed, 45 mm in diameter, and contained 3 cm$^2$ and 6 cm$^2$ wax paper channels positioned at the center and 1 mm above the fabric. All specimens were thermally conditioned in a room with 20.9 ± 0.1 °C and 55 % humidity for 24 h. Additionally, pre-conditioning the specimens and a 10-min cyclic compression (60% strain or 250 kPa, whichever achieved first) with lubricated specimens were performed (American Society for Testing and Materials, - ASTM, 2016). At the starting position, the compression plate rested on the specimen with a measured 0 ± 4 N contact force measured. This position constrained the channel's excessive expansion, similar to when the liner is actuated between the residual leg and socket. Each specimen was pressurized to 0, 10, or 20 psi immediately before running the test. The tubes from the control unit were clamped at the base of the specimen to prevent the backflow of $CO_2$ under compression. Each specimen was used once to discount the effect of compound plastic deformation. All the specimens were tested against leakage before and after the tests.

**Volumetric Elasticity:** The difference in the VE experiment setup compared to CE setup is the lateral restriction of the specimen while being compressed by the MTM (Mecmesin Multitest 5-xt, UK). This restriction is achieved by a custom-made fixture with a cavity with the specimen's exact diameter. Additionally, the compression plate has the same diameter as the cavity. In this setup, the specimen is compressed and not allowed to flow. Because of the incompressible nature of rubber and the inhibition of material flow by the cavity, VE experiments achieve higher stress at lower strain values compared to CE experiments.

### Liner Characterization
To measure the $F_A$, which is exerted perpendicular to the residual limb, a flexible silicone sheet containing the fluidic channels were placed flat between two aluminum plates (400 × 180 × 10 mm) and the zone in question was aligned with the center point of 1000 N loadcell (Mecmesin Multitest 5-xt, UK). The load cell was positioned and locked at 0.5 mm above the top steel plate to allow $CO_2$ to pass into the channels. Each zone was then gradually pressured 10, 15, and 20 psi, and the force generated was recorded. The test was conducted on each zone of Roliner and repeated on three liners.

We validated the key design requirement, robustness, before observing the force dynamics. The liner was constrained between two clear acrylic sheets (210 × 297 × 4 mm). By simulating liner's constrained position between the residuum and socket, this experiment

aimed to observe the maximum pressure in the channels before a layer or channel deformation. The tests were repeated with three liners and the average maximum pressure is 25 psi before a significant deformation occurred (Fig. 4).

The type of change in the pressure curves during the expansion of the channels can be characterized to map the gap between the residual limb and socket. This feature can be used for remote volume mapping and fitting assessment.

## Fluidics control
The next step was to design a manifold that can be both interfaced with our pressure source as well as microfluidics components (Valves: LHDA0531115H by The Lee Company, USA; Sensors: ABPDANN060P-GAA5 by Honeywell, USA). We designed a two-layer manifold that connects the pressure regulator to the valves and sensors assembled on the surface of the manifold. We used Stereolithography (SLA) to fabricate the manifold from EnvisionTec ABS Flex Black material at EnvisionOne cDLM Printer, which delivered a 50 μm layer resolution.

## Preclinical study
The inclusion and exclusion criteria of the study are provided in Supporting Information. There were no specific criteria for sex and/or gender. The age criterion was set to between ages of 18 and 60. The sex and/or gender, number and age of the participant along with their amputation and prosthetic details are provided in Table S7 and Table S8.

## Reporting summary
Further information on research design is available in the Nature Portfolio Reporting Summary linked to this article.

## Data availability
All data supporting the findings of this study are available within the article and its supplementary files. Any additional requests for information can be directed to, and will be fulfilled by, the corresponding authors. Source data are provided with this paper.

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

## Acknowledgements

F.G. would like to thank the following sponsors European Institute of Innovation and Technology Health (Reference: 19636 & 20671) for their generous support. U.T. also thanks UKRI – Innovate UK (Smart Grant 10004425 & 10032918) for their support. HK acknowledges the EU Horizon Europe Marie Skłodowska-Curie fellowship (Ref: 101111321) and UKRI MSCA fellowship (EP/Y030273/1) for their supports.

## Author contributions

U.T., G.S., T.A., D.T., and S.L.S. performed the experiments. U.T., G.S., T.A., and H.K. analyzed and visualized the results. U.T., G.S., T.A., H.K., J.S., L.B., A.M.J.B., and F.G. discussed the results. F.G. supervised the research. U.T., H.K., and F.G. wrote the manuscript. All authors reviewed the manuscript.

## Competing interests

U.T., G.S., T.A., and F.G. declare the existence of a financial competing interest as shareholders in Unhindr Ltd. The remaining authors declare no competing interests. U.T. and F.G. also declare the patent application (WO2019180431A1) for millifluidic integrated flexible sheets. U.T. declares the patent application (WO2021053350A1) for manufacturing method of millifluidic architecture. The remaining authors declare no conflict of interest.

## Ethics

Every experiment and questionnaire involving human participants have been carried out following a protocol approved by the Imperial College Research Ethics Committee (ICREC reference number: 21IC7315). Each participant gave informed written consent. The participant were joined the study voluntarily without any compensation.
