## [Transparent Peer Review file · Nature Communications]

Dynamically Adaptive Soft Metamaterial for Wearable Human-Machines Interfaces

Corresponding Author: Dr Firat Güder

Version 0:

Reviewer comments:

Reviewer #1

(Remarks to the Author)

Overall this research offers a unique solution to residual limb volume fluctuation issue in transfemoral prosthesis users. I have provided comments below.

P3L6: Suggest "regain" instead of "to gain back"

P3: suggest changing "shape of a residual leg" to "shape of residual limb"

P3: suggest "leading to a poor fitting"

P3: This sentence is misleading. True, a chronically poor fitting socket that is worn during physical activity may in fact lead to sores, ulcers, amputation. However, the beginning of your sentence misleads. I suggest "The residual limb dynamically changes its shape A prosthesis user wearing an ill fit prosthesis may experience residual limb sores, ulcers, and possible amputation."

P4: Change "sacrifice" to "alter"

P4: "Commercially available "passive" liners, made of silicone or polyurethane elastomers are used by amputees as a soft interface to improve comfort or suspension but not fitting." I'm not sure I understand this sentence. Liners certainly ensure comfort and suspension, but their function can't be disassociated with overall fit of the prosthesis. The prosthetist factors in the liner of choice into their plaster modification and definitive (hard) socket fabrication.

P5: "on demand by amputees" to "on demand by prosthesis users"

P5: First paragraph. I appreciate this added context for readers regarding liners.

P7: "To form a prosthetic liner that can be worn by the amputee" to "To form a prosthetic liner that can be worn by the prosthesis user"

P8: I recommend adjusting as such: "The fabric coating also reduced the tackiness of the surface of the liner making donning the prosthesis easier by the prosthesis wearer"

P16: "proven" to "can be supported"

P17: You recruited 6 participants but only evaluated 4? I see an "n=4".

P17: Should be "Which consisted of ten.."

Figure 6. For PEQ, are the ** indicating significant differences?

P19: Should be "in three of the four participants"

P19: This sentence is confusing and probably unnecessary "In contrast to gait analysis, fitting of a prosthetic liner is a subjective criterion."

P19: The PEQ is not a "clinically gold-standard" for PROMs. Its not an issue to use the term "gold-standard" in your liner comparison, but I do not recommend doing so for the PEQ.

P20: "For example, subject SJ04 had rapid weight loss of 20 kg prior to our pre-clinical studies. He did not have a socket replacement after losing weight therefore, he consistently experienced poor fitting everyday with his prosthetic socket. The gap between his residual limb and the socket was in the order of 1.5 cm, and the subject manually filled this gap by wearing additional layers of ply socks to walk." This is wonderful, however, a) His issue makes including him into your study and the

PEQ results potential a bias, b) if he was in effect “swimming” in his existing socket to begin with and in need of a new socket, what then did you use as the passive liner? If you used a liner and then socks this would certainly effect both gait and subjective performances and results.

P21: prove

P21: “to definitively prove the clinical efficacy of Roliner,” The use of the word ‘prove’ is not suggested.

Major criticisms:

Why did you choose transfemoral prosthesis users? Although they are a large group of the number of prostheses users, transtibial users are in greater need of devices.

I see no description of the passive liners you provided. Nor do I know how long these individuals were prosthesis users. For example SJ04, was already in an ill-fitting socket when you recruited and still used this socket for Roliner provision and subsequent data collection. Typically research prosthesis users are seasoned users of their device, and the device/s (interventions) are highly controlled in order to reduce potential bias.

I also don’t read anywhere details of prosthetic suspension types of these participants.

In the background you state “(in less developed regions, there is also a severe shortage of prosthetists that can fit sockets)”. I don’t understand your reference here. Surely there are likely a shortage of prosthetists whom know how to fabricate an adjustable type socket Martin or Otto Bock. And surely in resource limited environments there is limited funding which would put these adjustable sockets out of reach. Simply put, I don’t think you need to state “(in less developed regions, there is also a severe shortage of prosthetists that can fit sockets)”

You also state “Unlike conventional prosthetic sockets, Roliner does not require custom handmade manufacturing by a specialized technician,”

True. But it appears that it will require outsourced manufacturing by a specialized facility.

You state “Hence, we expect that these devices can be produced easily in most academic settings.” How can these devices be produced in most academic settings? And why would these devices need to produced in academic settings to begin with. The world of providing a prosthesis to a patient is already convoluted with visits to various rehab team members working together. Would a prosthetic clinic and their Prosthetist now need to outsource the liner provision process to an academic center?

The Roliner is incredible use of technology to address limb fluctuations. But probably by biggest concern is on the long-term limb health of the prosthesis user, outside a 6 trial 6 meter walking assessment. I would have liked to see a longer duration analysis in particular on limb health.

Moreover, suction type (ring) and elevated vacuum suspension systems in developed settings are typically utilizing a hydrostatic weight bearing type sockets. This entails the clinician to provide a socket which applies a negative pressure on the limb (refer to Carl Caspers, elevated vacuum). The rigid socket must be circumferentially intimately connected to the residuum, otherwise this vacuum could create a sucking of the skin which could burst open the skin wreaking havoc for the patient and prosthetist. This is especially concerning for the majority of amputees, whom are diabetic (dysvascular) amputees.

In viewing the videos and reading about Roliner adjustability. My concern is that I am unaware of how these Circular or Hexagonal patters will have long-term negative consequences on the limb under suction and/or elevated vacuum suspension. As an aside, both suction suspension and vacuum suspension are capable of mitigating volume fluctuations.

This statement: “Most importantly, Roliner will substantially improve health outcomes by reducing fitting sores and enhancing the quality of life for amputees as the main added value.” I’m unaware of a study evidencing reduced fitting sores by using Roliner.

Where was the expected 6-month lifespan estimate determined? The replacement of a CO2 cartridge every week may seem reasonable with a price under \$1. However this is one more thing that the prosthesis user needs to remember to do every week for the rest of their life. Also, will they be able to fly?

This last statement is kind of confusing. “Currently, most prosthetic sockets are handmade in clinics with long wait times;” Please define “long wait times”. And in which setting? The UK, U.S., Asia, etc. Government health clinic, private practice?

“hence, most amputees experience poor prosthetic fitting in their daily routines.”

How and why would long wait times result in a poor prosthetic fit during daily activities?

If you are trying to say that patient with ill-fitting sockets must continue to utilize their ill-fit socket whilst awaiting for a new

and better fit socket..Then okay I get this point. But if you aren't, then your statement is misleading.

Lastly, this an awesome technology with much potential. The clinical and third-party payer (insurance, L-Codes in USA, etc) can be worked out in due time. I could possible see the electro pneumatic control unit being somehow integrated into users of microprocessor type knee units as an added option. As they have some inherent capabilities which may work to serve two purposes.

Reviewer #2

(Remarks to the Author)

The authors present a soft robotic liner designed for transfemoral amputees. This liner's key feature is its ability to adapt to the socket and residual limb by selectively expanding specific zones. This expansion exerts targeted pressure on the residual limb, enhancing fit and comfort. Pre-clinical trials with amputees have shown that this soft robotic liner performs comparably to commercial liners.

- What is the maximum residual limb volume reduction the Roliner can handle? You can give it as a percentage.
- What is the estimated duration of the 16-gr canister during walking? This should be included in the results section before mentioning in the Conclusion.
- What is the estimated duration of the roliner in comparison to a passive liner?
- What is the silicone used to fabricate the roliner? Ecoflex 0050?
- What is the advantage of using hexagonal patterns compared to circular ones? Both geometries are shown in a video.
- How does the Roliner handle sweat and humidity, and what measures can be taken to ensure comfort and hygiene while wearing it?

This research is promising as pre-clinical trials suggest that the Roliner could potentially enhance dynamic fitting and comfort for lower limb amputees.

Version 1:

Reviewer comments:

Reviewer #1

(Remarks to the Author)

I appreciate your response to each of my comments. The manuscript provides a clearer understanding of each the approach, results and discussion. I look forward to this work translating to a clinical practice in the future.

Reviewer #2

(Remarks to the Author)

The authors have addressed all my concerns.

RESPONSE TO THE REVIEWER COMMENTS

Reviewer #1:

“Overall this research offers a unique solution to residual limb volume fluctuation issue in transfemoral prosthesis users. I have provided comments below.”

Our response: We would like to thank the reviewer for their time and thoroughness.

“P3L6: Suggest “regain” instead of “to gain back” ”

Our response: The “gain back” verb has been revised to “regain”. As the reviewer #1, it would be a better word selection.

“P3: suggest changing “shape of a residual leg” to “shape of residual limb” “

Our response: The “leg” wording has been revised to “limb”. It would better reflect the generalized use of prosthetics.

“P3: suggest “leading to a poor fitting” “

Our response: The “leading to poor fitting” term has been revised to “leading to a poor fitting”.

“P3: This sentence is misleading. True, a chronically poor fitting socket that is worn during physical activity may in fact lead to sores, ulcers, amputation. However, the beginning of your sentence misleads. I suggest “The residual limb dynamically changes its shape A prosthesis user wearing an ill fit prosthesis may experience residual limb sores, ulcers, and possible amputation.”

Our response: The sentence has been revised to read: “The residual limb dynamically changes its shape, leading to a poor fitting. A prosthesis user wearing an ill fit prosthesis may experience residual limb sores, ulcers, and possible further amputations”

“P4: Change “sacrifice” to “alter” “

Our response: The “sacrifice” verb has been revised to “alter” to better reflect the meaning of the sentence as reviewer #1 pointed out.

“P4: “Commercially available “passive” liners, made of silicone or polyurethane elastomers are used by amputees as a soft interface to improve comfort or suspension but not fitting.” I’m not sure I understand this sentence. Liners certainly ensure comfort and suspension, but their function can’t be disassociated with overall fit of the prosthesis. The prosthetist factors in the liner of choice into their plaster modification and definitive (hard) socket fabrication.”

Our response: The sentence has been revised to add more clarity which now reads as follows:
“Commercially available passive liners, made of silicone or polyurethane elastomers, are used by prosthetic

users as a soft interface to improve comfort, suspension, and fitting but cannot offer dynamic adjustment of these features.”

“P5: “on demand by amputees” to “on demand by prosthesis users” “

Our response: The “amputees” has been revised to “prosthesis users”.

P5: First paragraph. I appreciate this added context for readers regarding liners.

Our response: We would like to thank the reviewer #1 for his/her appreciation.

“P7: “To form a prosthetic liner that can be worn by the amputee” to “To form a prosthetic liner that can be worn by the prosthesis user” “

Our response: The “amputees” word has been revised to “prosthesis users”.

“P8: I recommend adjusting as such: “The fabric coating also reduced the tackiness of the surface of the liner making donning the prosthesis easier by the prosthesis wearer” ”

Our response: We revised the sentence as “The fabric coating also reduced the tackiness of the surface of the liner and made the donning of the prosthesis easier for the prosthesis wearer.”

“P16: “proven” to “can be supported” “

Our response: “proven” word has been revised to “supported”.

“P17: You recruited 6 participants but only evaluated 4? I see an “n=4”.”

Our response: Indeed as the reviewer correctly identified, we recruited six participants for the whole study, but only four prosthesis users participated in the gait analysis of Roliner in the motion capture system, hence $n = 4$ for those experiments. All six participants participated in the PROM study therefore they were included in Figure 6A as the cohort.

“P17: Should be “Which consisted of ten..” “

Our response: This change has been implemented as recommended.

*“Figure 6. For PEQ, are the ** indicating significant differences?”*

Our response: Yes, ** indicates statistical significance. To make this point clearer, we have added the following description into the caption of Figure 6: “(**) represents statistical significance as $p < 0.005$.”

“P19: Should be “in three of the four participants” “

Our response: “volunteer amputees” has been revised to “participants” as recommended.

“P19: This sentence is confusing and probably unnecessary “In contrast to gait analysis, fitting of a prosthetic liner is a subjective criterion.” “

Our response: The sentence has been removed for clarification as suggested.

“P19: The PEQ is not a “clinically gold-standard” for PROMs. Its not an issue to use the term “gold-standard” in your liner comparison, but I do not recommend doing so for the PEQ.”

Our response: The term “clinical gold standard” has been revised to “clinically accepted standard” .

“P20: “For example, subject SJ04 had rapid weight loss of 20 kg prior to our pre-clinical studies. He did not have a socket replacement after losing weight therefore, he consistently experienced poor fitting everyday with his prosthetic socket. The gap between his residual limb and the socket was in the order of 1.5 cm, and the subject manually filled this gap by wearing additional layers of ply socks to walk.” This is wonderful, however, a) His issue makes including him into your study and the PEQ results potential a bias, b) if he was in effect “swimming” in his existing socket to begin with and in need of a new socket, what then did you use as the passive liner? If you used a liner and then socks this would certainly affect both gait and subjective performances and results.”

Our response: Although it is an extreme case of ill fitting, it is still a probable use case. In fact, this extreme example shows that Roliner can replace the entire process of putting on a passive liner and using additional layers of socks to compensate of the volumetric mismatch.

- a) The participant used his own passive liner so his experience in study completely stemmed from his everyday experience of the ill-fit socket.
- b) He used his own ill-fit socket with additional socks as he does in his daily life. This had been the experience of this prosthesis user in his everyday life while waiting for a new fitting.

To show how the PEQ results are affected by removal of SJ04 from the statistics, we replotted the Fig 6c and presented as w/ and w/o SJ04 participant below.

including SJ04

excluding SJ04

Figure 1. The box chart comparison of Roliner versus passive liners in terms of prosthetic utility, ambulation, frustration and residual limb health parameters Prosthesis Evaluation Questionnaire (PEQ) scores from the preclinical cohort (except SJ04, removed for potential bias). The scores from each question response from each amputee are shown as black-filled circles. The horizontal line shows the median score. The mean scores (μ) and the normal distribution are shown on the right-hand side of the box chart. The lower and upper bounds of each box chart show 25th and 75th percentile, respectively. The lower and upper bounds of the whiskers show the 5th and 95th percentile, respectively. The p value calculated using paired sample t-test. (**) represents statistical significance as $p < 0.005$.

The removal of SJ04 data from PEQ statistics increased the mean values across the board for both conventional liners and Roliner. The distribution of the frustration and residual limb health results have a smaller spread as we exclude the SJ04 participant. The statistical significance was still well below $p < 0.005$ for all sections of PEQ analysis.

“P21: prove”

Our response: “prove” verb has been revised to “evaluate” as suggested.

“P21: “to definitively prove the clinical efficacy of Roliner,” The use of the word ‘prove’ is not suggested.”

Our response: “prove” verb has been revised to “evaluate” as suggested.

Major criticisms:

1) *Why did you choose transfemoral prosthesis users? Although they are a large group of the number of prostheses users, transtibial users are in greater need of devices.*

Our response: There were two main factors in deciding where to start the device development. The first one was that the area to work within a transtibial socket is much smaller. Placement of the seal, channel alignment, or integration of pin mechanism into liner (if shuttle lock liner were to be selected) would have been other design challenges in addition to the main design objective. The second factor was related to comfort issue. A transtibial liner is worn over the knee joint. The liner design (non-actuated area) at the back of the knee (the fold) could have introduced discomfort due to an accordion effect during knee flexion. This would have affected measuring the comfort of the adaptive liner. For these reasons, we have decided to first develop a version of Roliner for use with transfemoral prosthesis users.

2) I see no description of the passive liners you provided. Nor do I know how long these individuals were prosthesis users. For example SJ04, was already in an ill-fitting socket when you recruited and still used this socket for Roliner provision and subsequent data collection. Typically research prosthesis users are seasoned users of their device, and the device/s (interventions) are highly controlled in order to reduce potential bias.

Our response: We agree that this background information is important. We have not provided a passive liner ourselves, the participants brought their own daily liners to the experiments to simulate their daily experiences. We have, therefore, entered this information in addition to the length of prosthesis use for each participant in Table S3 in supporting information.

Table S1. The physical details of the participants in the study.

Subject Number	Sex	Age	Height (cm)	Weight (kg)	Amputation Side	Amputation Level	Prosthetic Use Experience (years)
SJ01	M	28	175	78	Bi-lateral	Through-knee	11
SJ02	M	29	192	77.8	Right Leg	Transfemoral	10
SJ03	M	43	197	115	Left Leg	Transfemoral	9
SJ04	M	34	183	107.9	Left	Transfemoral	15
SJ05	F	28	155	48	Left	Transfemoral	12
SJ06	M	58	172	76	Right	Transfemoral	11

3) I also don't read anywhere details of prosthetic suspension types of these participants.

Our response: The information for each participant is now included in supporting information, Table S4.

Table S2. The list of prosthetic limbs used by the participants in this study.

Subject Number	Prosthetic Limb Weight (kg)	Knee Model	Liner Brand	Liner Type	Liner Size	Removable Seal	Seal Size	Liner Weight (kg)	Liner Profile
SJ01	N/A	Stubbies	Ossur 4Seal	Seal-in	N/A	No	N/A	N/A	Conical
SJ02	5.0	Ossur Rheo Knee XC	Ossur Iceross	Seal-in	34	Yes	34	0.70	Cylindrical
SJ03	5.8	Ottobock Genium	Ossur Iceross	Seal-in	36	Yes	47	0.80	Conical
SJ04	5.4	C-leg 4 Otto Bock	Ossur Iceross	Seal-in	34	No	N/A	0.85	Conical
SJ05	4.2	C-leg 4 Otto Bock	Ossur Iceross	Seal-in	32	Yes	35	0.70	Conical
SJ06	4.8	Ottobock Genium	Ossur Iceross	Seal-in	38	No	N/A	0.80	Conical

4) *In the background you state “(in less developed regions, there is also a severe shortage of prosthetists that can fit sockets)”. I don’t understand your reference here. Surely there are likely a shortage of prosthetists whom know how to fabricate an adjustable type socket Martin or Otto Bock. And surely in resource limited environments there is limited funding which would put these adjustable sockets out of reach. Simply put, I don’t think you need to state “(in less developed regions, there is also a severe shortage of prosthetists that can fit sockets)”*

Our response: We agree. We removed the statement to avoid confusion.

5) *You also state “Unlike conventional prosthetic sockets, Roliner does not require custom handmade manufacturing by a specialized technician,”*

Our response: True. But it appears that it will require outsourced manufacturing by a specialized facility. Indeed, Roliner would require outsourced manufacturing yet be suitable for mass production. The predetermined size of the Roliner can be used by prosthesis user and the gap fit compensation can be remedied with dynamic pressurization of the Roliner.

6) *You state “Hence, we expect that these devices can be produced easily in most academic settings.” How can these devices be produced in most academic settings? And why would these devices need to be produced in academic settings to begin with. The world of providing a prosthesis to a patient is already convoluted with visits to various rehab team members working together. Would a prosthetic clinic and their Prosthetist now need to outsource the liner provision process to an academic center?*

Our response: We tried to stress that these types of dynamic devices can be prototyped in most academic settings, especially important for research and partially for small scale production in academic labs for trials: we hope that others in the academic community will build on our study and improve the device and perhaps reengineer for different problems that we have not addressed in this work. We revised the statement as “Hence, we expect that these devices can be prototyped in most academic settings.”

7) *The Roliner is incredible use of technology to address limb fluctuations. But probably by biggest concern is on the long-term limb health of the prosthesis user, outside a 6 trial 6 meter walking assessment. I would have liked to see a longer duration analysis in particular on limb health.*

Our response: In this work, we focused primarily on the engineering of Roliner and preliminary pre-clinical trials to demonstrate viability of the technology. We applied for ethics for a larger, national trial where we will conduct a 9-week trial with 14 participants but of course this study will take a lot longer, therefore, we have decided to publish without much more delay with the preliminary trials to share our results with the wider community and public to receive feedback.

8) *Moreover, suction type (ring) and elevated vacuum suspension systems in developed settings are typically utilizing a hydrostatic weight bearing type sockets. This entails the clinician to provide a socket which applies a negative pressure on the limb (refer to Carl Caspers, elevated vacuum). The rigid socket must be circumferentially intimately connected to the residuum, otherwise this vacuum could create a sucking of the skin which could burst open the skin wreaking havoc for the patient and prosthetist. This is especially concerning for the majority of amputees, whom are diabetic (dysvascular) amputees.*

Our response: Roliner pressurizes its channels to expand and compensate for the possible misfitting between the residuum and the rigid socket. This expansion helps maintaining the intimate fit as the reviewer highlighted). As a result of this pressurization, Roliner acts like a compression sock on the skin, but also the expansion pushes the seal closer to the socket to fill the gaps and ensure a sufficient vacuum.

9) *In viewing the videos and reading about Roliner adjustability. My concern is that I am unaware of how these Circular or Hexagonal patterns will have long-term negative consequences on the limb under suction and/or elevated vacuum suspension. As an aside, both suction suspension and vacuum suspensions can mitigate volume fluctuations.*

Our response: Roliner expands in two directions: towards the skin and towards the socket. This behaviour creates a circumferential compression effect. We have not investigated the long-term implications of circular/hexagonal patterns in this study. It is known that volume compensating systems could affect the volume of the residual limb over long term.¹ We plan to investigate the long-term effects of hexagonal patterns in a 9-week long national clinical investigation, that is in submission process.

¹ Sanders, J. E., Harrison, D. S., Allyn, K. J., Myers, T. R., Ciol, M. A. & Tsai, E. C. (2012) How do sock ply changes affect residual-limb fluid volume in people with transtibial amputation? *Journal of Rehabilitation Research and Development; J Rehabil Res Dev.* 49 (2), 241–256. 10.1682/JRRD.2011.02.0022.

10) *This statement: “Most importantly, Roliner will substantially improve health outcomes by reducing fitting sores and enhancing the quality of life for amputees as the main added value.” I’m unaware of a study evidencing reduced fitting sores by using Roliner.*

Our response: We revised this line to add more clarity which now reads: “Most importantly, Roliner will likely enhance the quality of life for prosthesis users.”.

11) *Where was the expected 6-month lifespan estimate determined? The replacement of a CO2 cartridge every week may seem reasonable with a price under \$1. However this is one more thing that the prosthesis user needs to remember to do every week for the rest of their life. Also, will they be able to fly?*

Our response: Conventional passive liners can last 6-9 months of use. We estimated the Roliner's lifetime conservatively at 6 months since both conventional liners and Roliner are made of the same type of material, i.e., silicone elastomers. We opted to remove this estimate as we do not have definitive evidence at the current stage of development. We hope to have a definitive lifespan after longer trials.

We have carefully investigated the potential issue concerning the transport of gas cartridges on passenger flights. International Air Transport Association (IATA) permits that non-flammable, non-toxic gas cylinder worn for the operation of mechanical limbs can be carried onto an airplane. The prosthesis users using Roliner would also not need to seek any prior approval of the operator or inform the pilot-in-command as listed in IATA guidelines (Provisions for dangerous goods carried by passengers or crew, Table 2.3.A – Subsection 2.3).¹ In fact, similar gas cylinders are also used in inflatable life jackets.

¹ IATA, Dangerous Good Regulation, Provisions for Dangerous Goods Carried by Passengers or Crew. <https://www.iata.org/contentassets/6fea26dd84d24b26a7a1fd5788561d6e/dgr-62-en-2.3a.pdf>, retrieved on October 24th, 2024.

12) *This last statement is kind of confusing. “Currently, most prosthetic sockets are handmade in clinics with long wait times;”*

Please define “long wait times”. And in which setting? The UK, U.S., Asia, etc. Government health clinic, private practice?

Our response: We revised the statement to add more precision which now reads: “Currently, most prosthetic sockets are handmade in clinics with long wait times (3-4 weeks in UK and US);”^{1,2}

¹ Olsen, J, Day, S, Dupan, S, Nazarpour, K & Dyson, M 2021, '3D-Printing and upper-limb prosthetic sockets: promises and pitfalls', IEEE Transactions on Neural Systems and Rehabilitation Engineering, vol. 29, pp. 527-535. <https://doi.org/10.1109/TNSRE.2021.3057984>

² Turner S, Belsi A, McGregor AH. Issues Faced by Prosthetists and Physiotherapists During Lower-Limb Prosthetic Rehabilitation: A Thematic Analysis. Front Rehabil Sci. 2022 Jan 10;2:795021. doi: 10.3389/fresc.2021.795021. PMID: 36188791; PMCID: PMC9397966.

13) *“hence, most amputees experience poor prosthetic fitting in their daily routines.”
How and why would long wait times result in a poor prosthetic fit during daily activities?*

Our response: We agree that this statement can cause some confusion, therefore, we have decided to remove it as the issues around fitting have already been described elsewhere in the manuscript in sufficient detail.

14) *If you are trying to say that patient with ill-fitting sockets must continue to utilize their ill-fit socket whilst awaiting for a new and better fit socket. Then okay I get this point. But if you aren't, then your statement is misleading.*

Our response: Indeed this is what were trying to convey, however, we already mentioned that point more generally elsewhere with more clarity, hence we decided it is a good idea remove this.

15) *Lastly, this an awesome technology with much potential. The clinical and third-party payer (insurance, L-Codes in USA, etc) can be worked out in due time. I could possible see the electro pneumatic control unit being somehow integrated into users of microprocessor type knee units as an added option. As they have some inherent capabilities which may work to serve two purposes.*

Our response: We were absolutely delighted to hear this. Thank you! We also envision that the control unit of Roliner can be integrated into robotic limbs monolithically to create a better experience and reduce costs further through elimination of redundant systems (microcontroller, batteries etc.).

Reviewer #2:

The authors present a soft robotic liner designed for transfemoral amputees. This liner's key feature is its ability to adapt to the socket and residual limb by selectively expanding specific zones. This expansion exerts targeted pressure on the residual limb, enhancing fit and comfort. Pre-clinical trials with amputees have shown that this soft robotic liner performs comparably to commercial liners.

1) What is the maximum residual limb volume reduction the Roliner can handle? You can give it as a percentage.

Our response: We investigated the maximum suspension force with a 5 mm and 10 mm gap with Roliner in Figure 4D. In the experiment, the non-actuated Roliner (device under test) diameter was 95 mm, and we successfully reached a suspension force of 0.5 kN with both the gaps of 5 and 10 mm. The compensated volume percentage can be calculated as -7.8% (shrinkage) and -15.1% (shrinkage) for the gaps of 5 and 10 mm, respectively. A 10 mm gap is already considered to be quite large in the context of fitting of prosthetic sockets.

2) What is the estimated duration of the 16-gr canister during walking? This should be included in the results section before mentioning in the Conclusion.

Our response: A single 16-gr CO₂ canister can provide 2000 actuations for Roliner. The canister's use duration can be over a week (p21, line 4); however, it would also depend on how frequently the user tweaks their Roliner. As the reviewer suggests, we added this information to p16: "This experience can be avoided by pulsed actuation, which would also save pressurization gas from the portable canister, allowing longer-lasting usage up to 2000 actuations."

3) What is the estimated duration of the roliner in comparison to a passive liner?

Our response: The conventional passive liners can last 6-9 months of use. We estimated conservatively Roliner use lifetime as 6 months since both conventional liners and Roliner are made of out of same type of material, i.e silicone elastomers.

4) What is the silicone used to fabricate the roliner? Ecoflex 0050?

Our response: Both cylindrical fluidic body and dome of the Roliner is made out of Ecoflex OO-50. We also added the type of silicone into the SI (p1 line 3).

5) What is the advantage of using hexagonal patterns compared to circular ones? Both geometries are shown in a video.

Our response: We wanted to demonstrate different geometries can be achieved, however, due to higher packing density, we opted for hexagonal patterns. Hexagonal pattern also provided a more planarly uniform expansion. Additionally, the circular channels have a larger surface area where the silicone can expand more (like a dome). This expansion could cause the material to catastrophically burst under certain conditions.

6) How does the Roliner handle sweat and humidity, and what measures can be taken to ensure comfort and hygiene while wearing it?

Our response: Since the Roliner is manufactured from the same materials as conventional liners, similar sweat and humidity conditions apply. Roliner can be hand-washed daily to ensure comfort and hygiene. A perforated liner design can also be implemented to improve breathability, however, we have not pursued that avenue of development since conventional liner developers already patented the perforated liners.^{1,2}

¹ Silcare Breathe™ Walk (perforated liner), <https://www.blatchfordmobility.com/en-gb/products/liners/silcare-breathe-walk-perforated-liner> , retrieved on October 24th, 2024.

² D. Walter, Perforated liner, US Patent 10639173B2, <https://patents.google.com/patent/US10639173B2/en>, 2015

“This research is promising as pre-clinical trials suggest that the Roliner could potentially enhance dynamic fitting and comfort for lower limb amputees.”

Our response: We thank the reviewer for their valuable time and comments.